# Increasing proportion of mildly aged population in rural mitigates farmland abandonment in the farming-pastoral ecotone of northern China

Yuling Jin[1], Guoliang Zhang[2], Xin Chen[3], Yi Zhou[4], Yukai Wei[5], Sicheng Mao[6], Haile Zhao[1], Wenting Liu[1], Zhihua Pan[7], Pingli An [1]¤*

1 College of Land Science and Technology, China Agricultural University, Beijing, China, 2 Jinan Academy of Social Sciences, Jinan, China, 3 Institute of Loess Plateau, Shanxi University, Taiyuan, China, 4 School of Geographical Sciences, Hunan Normal University, Changsha, China, 5 Shandong Territorial Spatial Planning Institute, Jinan, China, 6 Bank of Communications Co., Ltd. Shaanxi Branch, Xi'an, China, 7 College of Resources and Environmental Science, China Agricultural University, Beijing, China

¤ Present address: College of Land Science and Technology, China Agricultural University, No. 2 Yuanmingyuan West Road, Haidian, Beijing 100193, China.
* anpl@cau.edu.cn

## Abstract

Rural population aging has emerged as a widespread phenomenon, which can lead to farmland abandonment and pose unprecedented challenges to agricultural production and ecological sustainability in the farming-pastoral ecotone of northern China (FPENC). The dynamic changes in farmland abandonment from 2000 to 2020 were systematically explored using a trajectory-based land use change detection approach. Binary logit regression models were employed to analyze the driving mechanism of the current farmland abandonment based on 1,195 questionnaires, and then random forest models were used to predict future farmland abandonment trends under various scenarios. The results showed that (1) rural population aging had emerged as a significant challenge for Ulanqab, with the proportion of people aged 60 and above increasing from 12.3% in 2000 to 44.9% in 2020. Members of mildly aged households (60–69) were identified as the main agricultural labor force, accounting for 46.6%; (2) an overall downward trend of farmland abandonment was observed, decreasing from 295 km² in 2000 to 273 km² in 2020. However, the abandonment rate increased slightly from 3.03% to 3.66%, with higher abandonment rates concentrated in northern Ulanqab; (3) an increase in the proportion of the mildly aged population mitigated farmland abandonment, while an increase in the proportion of the severely aged population (≥70) exacerbated it. When other conditions remained unchanged, a 5% and 10% increase in the proportion of the mildly aged population corresponded to a decrease in farmland abandonment rates to 11.4% and 10.5%, respectively; (4) the mechanisms underlying abandonment behavior differed between young and elderly households. For each additional year of age for elderly households, the probability of farmland abandonment increased by 4.1%.

**Data availability statement:** Some of the publicly available data utilized in this study can be accessed and downloaded from the following sources: Land use data is available from the zenodo database: (https://doi.org/10.5281/zenodo.4417809). The cropland data is available from the zenodo database: (https://doi.org/10.5281/zenodo.7936885). The SSPs data is available from the zenodo database: (https://zenodo.org/records/4554571). The questionnaire data contain qualitative responses from local farmers, which may indirectly disclose identifiable information about specific villages or communities, raising potential privacy concerns. During the research process, we informed village leaders and farmers that all data would be treated with strict confidentiality. As a result, the datasets generated and analyzed during the current study are not publicly available due to confidentiality agreements and the sensitivity of the data sources. Researchers interested in accessing the data for academic purposes are encouraged to contact the authors to discuss potential data sharing under appropriate conditions and with a formal data use agreement. We guarantee that the following contact information will remain valid and can be used for long-term correspondence and to respond to data access requests. Contact for data requests: [Shuai Wang] [wangshuai2019@cau.edu.cn] [+010-62733168] College of Land Science and Technology, China Agricultural University [Yuling Jin] [jinyuling@cau.edu.cn] College of Land Science and Technology, China Agricultural University.

**Funding:** This work was supported by the National Natural Science Foundation of China (grant number 42271268).

**Competing interests:** The authors have declared that no competing interests exist.

Elderly households with higher levels of education, fewer farm laborers, smaller per capita farmland area, a higher proportion of dryland, larger quantities of farmland, and non-contracted farmland were more likely to abandon farmland; (5) under three Shared Socioeconomic Pathway scenarios (SSP1, SSP2, and SSP5), the proportion of abandoned farmland by farmers was projected to rise to 58.8%, 20.1%, and 58.8%, respectively. These findings provided new insights into the issue of abandoned farmland, particularly from a demographic perspective.

## Introduction

Farmland abandonment is recognized as a prevalent type of land-use change globally and refers to a direct manifestation of farmland marginalization [1]. Previous studies have indicated that from 1992 to 2020, the global area of abandoned farmland reached 101 million hectares [2], with abandonment observed to varying degrees across Europe [3,4], North America [5], and Asia [6,7]. However, in China, with the rapid advancement of industrialization and urbanization, the large-scale outflow of agricultural labor and the continuous decline in agricultural income have contributed to the ongoing expansion of abandoned farmland, which has become progressively more prominent and severe [8]. Abandoned farmland has affected rural livelihoods and agricultural practices [9], and also impacted ecosystem structures [10] and biodiversity patterns [11,12], thereby exerting significant impacts on socio-economic development and the ecological environment [13,14]. As a populous country and a major agricultural producer, the abandonment of farmland in China has been recognized as a potential threat to national food security [15,16]. Therefore, there is an urgent need to address this issue and to formulate context-specific mitigation strategies.

Farmland abandonment is a complex process influenced by the interaction of multiple factors, including natural conditions, location conditions, the level of economic development, etc. [16–18]. Among these factors, population aging has increasingly been recognized as a critical influencing variable [19]. With declining birth rates and increasing life expectancy, population aging has become an increasingly serious social issue [20], particularly in rural China where it is more pronounced [21]. According to the Seventh National Population Census in 2020 in China, the population aged 60 and above was approximately 264 million, accounting for 18.7% of the total population. In rural areas, 29.4% of agricultural workers fell into this age group. The rising proportion of elderly individuals, along with diminished labor capacity and declining willingness to manage farmland, has intensified farmland abandonment [19]. However, the relationship between aging and farmland abandonment remains contested. Some studies have established a positive association between farmers' age and farmland abandonment in rural areas, particularly in mountainous regions [22–26], while others have found a negative correlation between the age group of 65 and over and the rate of abandonment [27]. These contradictory findings underscore the need for mechanism identification, particularly in region-specific contexts within China, where the pathways of influence and spatial heterogeneity require further investigation.

Traditional methods of investigating farmland abandonment have included field surveys, literature reviews, and other approaches to analyze the current situation and causes of farmland abandonment [14,16,22]. However, some limitations persist, such as the difficulty in reflecting the overall spatial pattern of farmland abandonment. Farmland abandonment is not a static but rather a dynamic process [28]. Currently, the abundance of satellite datasets provides a feasible data source for monitoring the spatio-temporal variation in farmland abandonment and assessing its characteristics [18,29–31]. Therefore, the effective integration of macro-remote sensing data with micro-questionnaire data could lead to a more comprehensive understanding of the driving mechanisms behind farmland abandonment. Additionally, machine learning could model with high precision based on the law of data [32]. This study employed machine learning models to explore the relationship between driving factors and farmland abandonment, establish a farmland abandonment prediction model, and analyze the future trend of farmland abandonment.

The farming-pastoral ecotone of northern China (FPENC) is recognized as a highly sensitive ecoclimatic region characterized by high land use intensity and ecological fragility [33,34]. Since the 20th century, the FPENC has also experienced more pronounced and severe rural population aging compared to other areas [35]. Compared with previous studies on farmland abandonment [36–38], the ecological vulnerability and accelerating population aging in the FPENC underscore the importance of placing greater emphasis on balancing ecological productivity and land use in addressing farmland abandonment. Therefore, in-depth analyses of the impacts of farmland abandonment in the FPENC with deep population aging are crucial for ensuring regional food security and ecological conservation, and can also provide valuable insights for similar regions in China and globally.

In this study, we chose Ulanqab in the central part of the FPENC, which is characterized by pronounced population aging and obvious phenomenon of abandoned farmland, as a typical research area. Based on our previous research [26,39], a theoretical framework was developed to analyze in depth the impact of rural population aging on farmland abandonment. The principal research objectives are as follows: (a) to analyze the spatio-temporal distribution and characteristics of farmland abandonment from 2000 to 2020; (b) to explore the mechanism of rural population aging on farmland abandonment across different household age groups; (c) to predict the changing trends of farmland abandonment in the context of population aging.

## The theoretical framework and research hypotheses

Drawing upon the theory of economies of scale in land use and the law of diminishing returns [40], household farmland-use behavior can be interpreted as a rational decision aimed at maximizing benefits under constraints of resource endowment [22,41]. When the scale of household farmland reaches an optimal threshold ($A_2$), per capita net income is maximized. If the farmland area is below the optimal threshold ($A<A_2$), income maximization has not been achieved, and households tend to expand their farmland to improve income. Conversely, if the farmland area exceeds the optimal threshold ($A>A_2$), households tend to sublet or abandon farmland to reduce operational scale and restore optimal efficiency. For farmland with low subletting value or poor quality, abandonment is often considered inevitable (Fig 1).

However, in practice, most households do not exhibit fully rational behavior. According to the "Limited Rational Hypothesis" proposed by Herbert A. Simon, individuals generally possess only "limited rationality" and tend to seek "satisfactory" rather than "optimal" outcomes in decision-making [42]. In this context, household characteristics, particularly age structure, significantly influence their farmland use decisions, thereby contributing to differences in the driving mechanisms of farmland abandonment. Therefore, households are classified into young households (farming and below 60 years old) and elderly households (farming and above or equal to 60 years old) by age and farming status (Fig 2a) [43–45].

The moderate scale of farmland management represents the optimal allocation of various production factors, such as land, labor, capital, and technology [46,47]. Compared to young households, elderly households generally exhibit reduced physical strength and economic inputs, and they experience a diminished capacity for learning and lower adoption of technology [22,48]. These constraints collectively reduce the optimal farmland management scale for elderly farmers

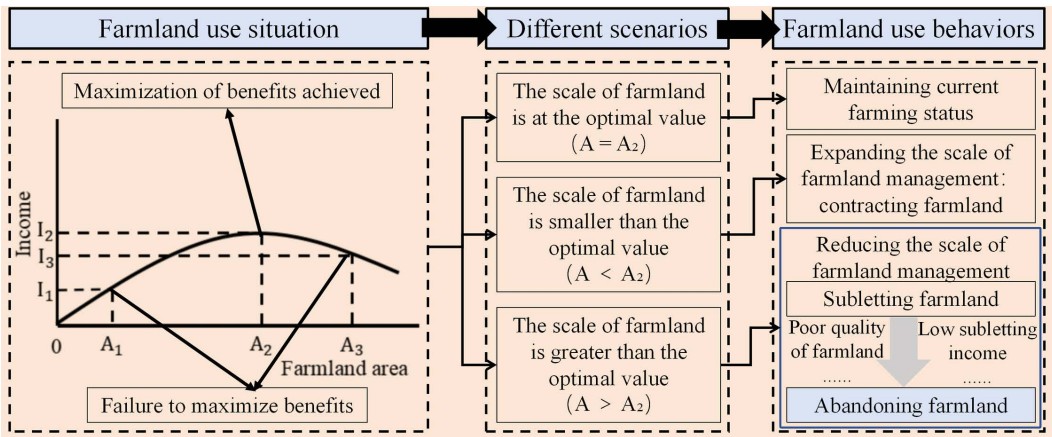

**Fig 1. Theoretical analysis framework for household farmland-use behavior.**

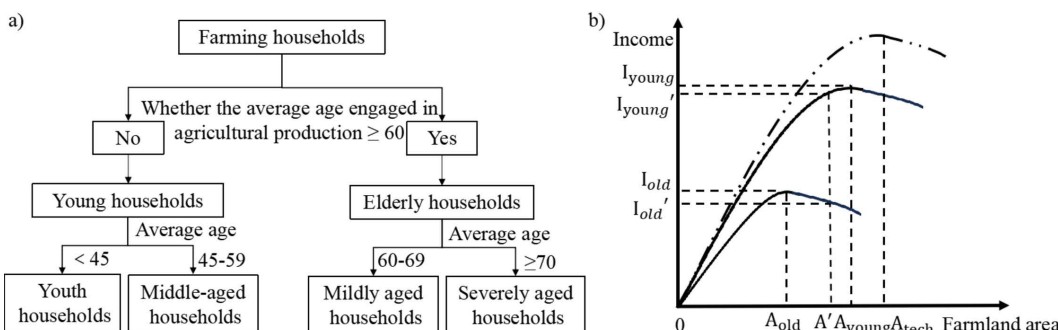

**Fig 2. Household type and farmland abandonment mechanism. (a)** Classification of farming household types by age. **(b)** Analysis of the farmland abandonment mechanism under age-related changes. Specifically, as technology continues to advance, the optimal farmland operation scale ($A_{tech}$) also increases, as represented by the dashed curve.

($A_{old}$), which is lower than that of younger households ($A_{young}$). When the actual farmland area for elderly farmers exceeds the optimal scale, it becomes difficult to achieve maximum agricultural income and thereby increasing the risk of farmland abandonment (Fig 2b). Furthermore, many elderly households exhibit a low willingness to continue farming due to limited agricultural income and high labor intensity. Therefore, abandonment often becomes a passive and unavoidable choice. This study proposes Hypothesis 1: age will exacerbate farmland abandonment in elderly households.

With the progression of population aging, elderly members often lose the physical capacity to continue farming, prompting the transfer of farmland management responsibilities to young members. However, these younger members face considerable opportunity costs: compared to the relatively low returns from cultivating farmland, urban employment offers more attractive economic prospects. Consequently, young households are more inclined to abandon farmland. Therefore, the following research hypothesis is proposed:

Hypothesis 2: the driving factors of farmland abandonment differ significantly across different age groups of households.

Furthermore, elderly households are not a homogeneous group and should be further differentiated. Although some households have entered the aging stage, they retain a degree of agricultural production capacity due to relatively good

physical health and available labor. These are referred to as "mildly aged households" (Fig 2a). They choose to return to rural areas for farming because of limited urban employment opportunities [49], and may maintain existing farmland use or even continue agricultural production through small-scale reinvestment. In contrast, severely aged households are more likely to abandon farmland due to diminished labor capacity, and a lack of motivation or ability to engage in farming activities. Therefore, this study proposes Hypothesis 3: an increase in the proportion of the mildly aged population will alleviate farmland abandonment to some extent.

## Materials and methods

### Study area

Ulanqab lies between 39°37'~43°28'N and 109°16'~114°49'E in the central part of the FPENC, located in northern China (Fig 3a). The area is characterized by a semiarid temperate continental monsoon climate, spanning approximately 54,500 square kilometers, and comprising 11 counties or banners (Fig 3b). The mean annual precipitation ranges from 150 to 450 mm, with 60% to 70% occurring in summer and only 2% to 3% in winter. Ulanqab represents a mosaic that serves as a transition zone between traditional farming and pastoral regions, situated within a single cropping system area [50]. In 2022, the planted area of crops reached 689,000 hectares, reflecting an increase of 4.1%, while the planted area of grain crops totaled 467,000 hectares, including 146,000 hectares for maize and 81,000 hectares for oats. It has witnessed a decrease in the cultivation area of wheat and potatoes and a sharp increase in the cultivation area of maize, oats, and soybeans [51]. However, the share of the primary industry in GDP in Ulanqab, which includes agriculture and animal husbandry, decreased from 54.58% to 16.51% between 1990 and 2021 [52].

Nowadays, rural population aging has emerged as a significant challenge for Ulanqab [53]. The aging coefficient (the proportion of people aged 60 and above to the total resident population) in rural Ulanqab has consistently exceeded that of Inner Mongolia and China since 2000, steadily increasing to approximately 44.9% by 2020 (Fig 4a). Additionally, a significant decline in the youth population (−19.6%) in rural Ulanqab was observed between 2010 and 2020 (Fig 4b). The degree of rural aging in Ulanqab has exhibited regional variations, with a low central region surrounded by a high

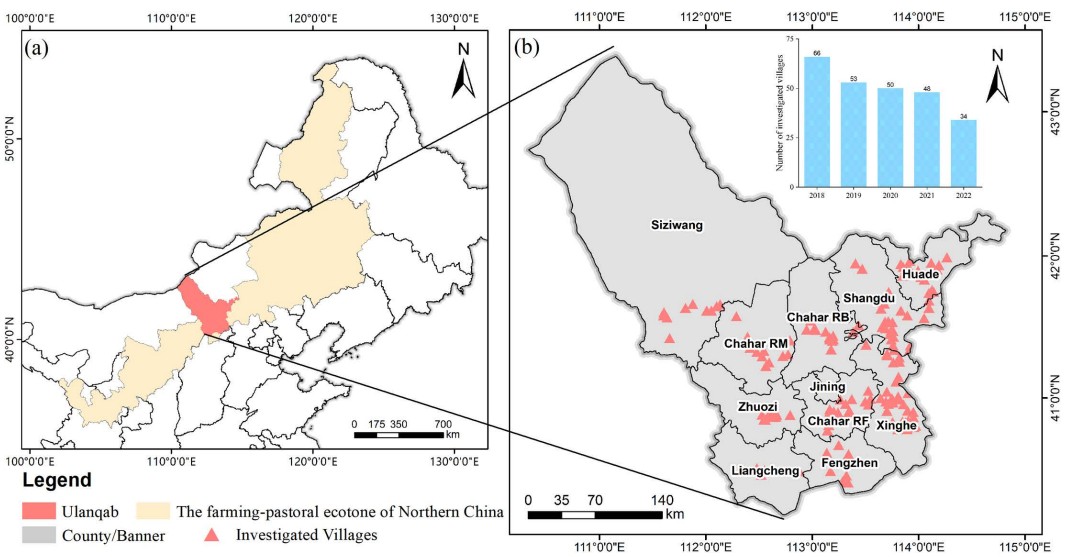

**Fig 3. Overview of the study area. (a)** The location of Ulanqab in China and the FPENC. **(b)** The distribution of survey villages and the number of questionnaires in Ulanqab. All map boundary data in this figure are consistent and publicly available from the National Platform for Common Geospatial Information Services (www.tianditu.gov.cn). The review map number is GS (2024) 0650.

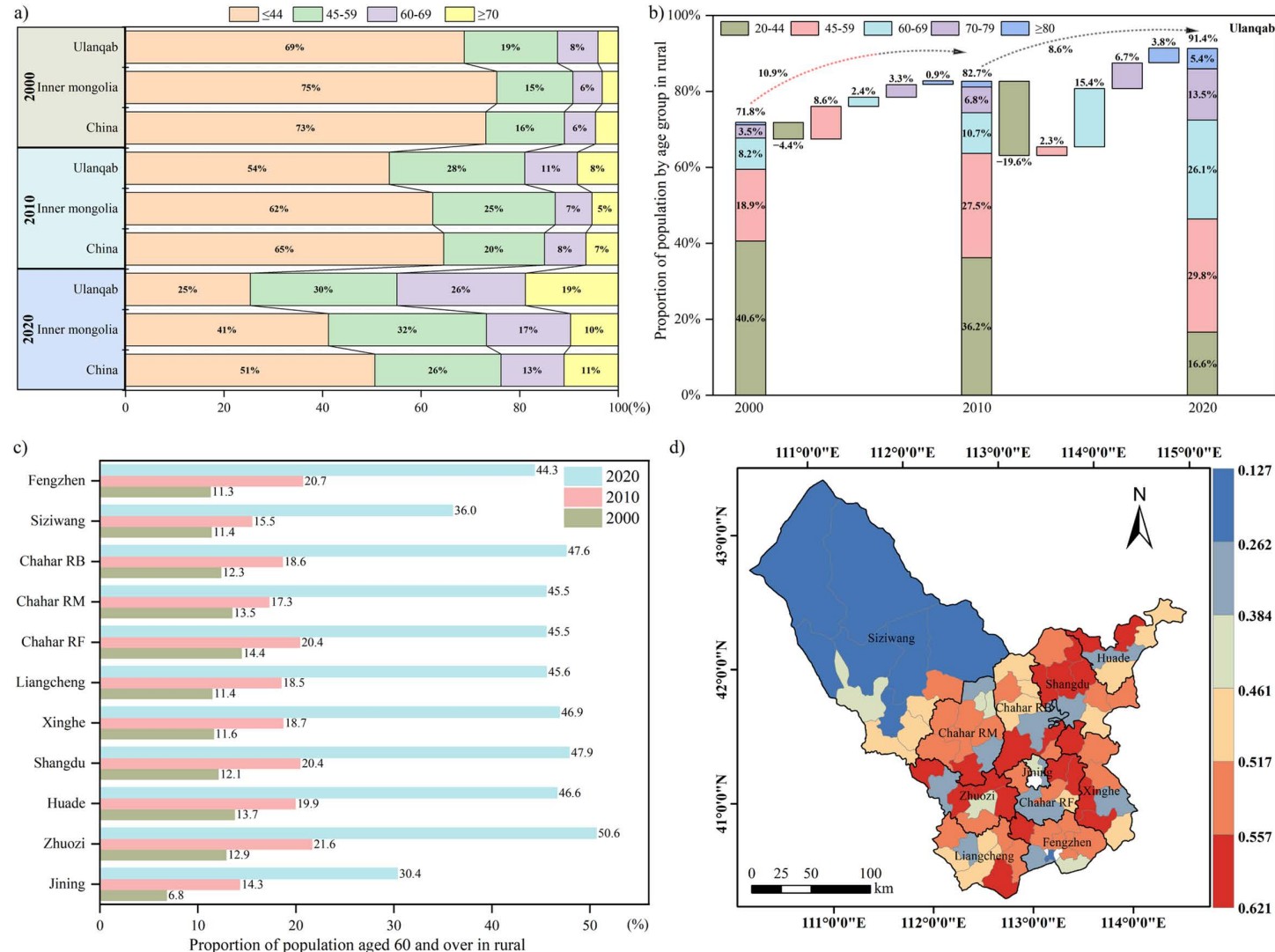

**Fig 4. Situation of the rural population aging in Ulanqab during 2000-2020. (a-b)** Composition of the rural population of China, Inner Mongolia and Ulanqab by age groups. **(c)** The rural population aging coefficient of counties and banners in Ulanqab. **(d)** The rural population aging coefficient of villages and towns in 2020. All map boundary data in this figure are consistent and publicly available from the National Platform for Common Geospatial Information Services (www.tianditu.gov.cn). The review map number is GS (2024) 0650.

perimeter. Specifically, rural areas dominated by cultivation (e.g., Xinghe) have shown a higher aging coefficient compared to those dominated by pastoralism (e.g., Siziwang Banner). In 2020, 74.1% of villages and towns recorded an aging coefficient exceeding 40%, highlighting the severity of the aging issue in these areas (Fig 4c-4d).

## Data and processing

**Questionnaire data.** The collection of data and information for this study was conducted over five periods. In the first period (July 15–30, 2018), a team of 6 individuals conducted the farming household survey, collecting 281 questionnaires, of which 254 were valid. In the second period (August 2–11, 2019), a team of 7 individuals carried out the farming household survey. A total of 234 farming household questionnaires were collected, with 229 valid. In the

third period (August 18–26, 2020), a team of 6 individuals conducted the third farming household survey. They collected 224 questionnaires, of which 216 were valid. In the fourth period (July 22-August 2, 2021), a team of 8 individuals conducted the fourth farming household survey. They collected 257 questionnaires, with 252 considered valid. In the fifth period (July 27-August 4, 2022), a team of 9 individuals, conducted the fifth farming household survey. They collected 262 questionnaires, of which 244 were valid. The questionnaire data involved farming household characteristics, farmland conditions, inputs and outputs, farmland abandonment, and willingness of households to cultivate in the future, among other aspects. A total of 1258 questionnaires were collected, with 1195 were valid, and the effective rate of the questionnaire was 95.0%. The research area encompassed 11 banners or counties, 57 towns, and 239 villages within Ulanqab (Table 1).

**Land use data.** Land use data with a 30m resolution from 1998 to 2022 was collected from the multi-year land cover dataset of China (CLCD)(https://doi.org/10.5281/zenodo.4417809) to facilitate the extraction of farmland abandonment. The CLCD offers a higher spatial resolution and longer temporal coverage compared to existing annual products, with an overall accuracy rate of 79.31% [54]. The extraction of abandoned farmland depends on the accuracy of farmland in land use classification, therefore, farmland data with a 30m resolution from 1998 to 2021 were collected from the cropland dataset of China (https://doi.org/10.5281/zenodo.7936885) to assist in auxiliary extraction [55].

**Auxiliary data.** Rural population data at the national level, as well as for Inner Mongolia and Ulanqab, for 2000, 2010, and 2020, were collected from the National Bureau of Statistics and the Ulanqab Bureau of Statistics [53]. The future population grid data were collected from population grid data for SSPs from 2020 to 2100. These data were constructed within the global framework of Shared Socio-economic Pathways (SSPs) and were used to generate provincial population details, including age, sex and educational attainment, by applying the recursive multidimensional model [56]. Furthermore, the slope data derived from the SRTMGL1 v003 at a 30m spatial resolution were used to exclude areas associated with Grain for Green Project (https://lpdaac.usgs.gov/products/srtmgl1v003/).

## Methods

**Extraction of farmland abandonment.** The definition of farmland abandonment remains inconsistent within the scientific literature [15]. According to the definition provided by the FAO, farmland that has been without agricultural production for 5 years or more is defined as abandoned [57]. Therefore, a five-year time window was established to identify abandoned farmland by using the land use trajectory method. Specifically, if a pixel is classified as farmland in the

**Table 1. Areas and villages included in the questionnaire survey data collection.**

| County/Banner | Village/Town | Number | Ratio(%) |
|---|---|---|---|
| Jining | (1): Baihaizi | 7 | 0.59 |
| Zhuozi | (4): Bayinxile, Zhuozishan, Shibatai, Dayushu | 104 | 8.70 |
| Chahar Right Front Banner | (6): Tuguiwula, Pingdiquan, Bayintala, Huangqihai, Meiguiying, Sanchakou | 98 | 8.20 |
| Fengzhen | (5): Longshengzhuang, Hongshaba, Jubaozhuang, Nanchengqu, Guantunbao | 86 | 7.20 |
| Xinghe | (8): Chenguan, Saiwusu, Eerdong, Dakulian, Minzutuanjie, Wuguquan, Datongyao, Wuguquan | 202 | 16.90 |
| Liangcheng | (5): Hongmao, Liusumu, Tiancheng, Maihutu, Daihai | 92 | 7.70 |
| Chahar Right Back Banner | (6): Baiyinchagan, Benhong, Honggeertu, Daliuhao, Tumuertai, Wulanhada | 122 | 10.21 |
| Chahar Right Middle Banner | (4): Kebuer, Huangyangcheng, Guangyilong, Hongpan | 128 | 10.71 |
| Shangdu | (8): Qitai, Shibaqing, Daheishatu, Tunkendui, Xiaohaizi, Dakulun, Bolihujing, Sandaqing | 200 | 16.74 |
| Huade | (5): Changshun, Chaoyang, Debaotu, Gonglahudong, Baiyintela | 99 | 8.28 |
| Siziwang Banner | (5): Wulanhua, Jishengtai, Kuluntu, Dongbahao, Hujitu, | 57 | 4.77 |
| Total | 57 | 1195 | 100.00 |

base year, we define it as "abandoned farmland" if it becomes grassland or bare land in the test year and remains in this state for the subsequent three years [58]. Since the conversion of farmland to grassland in the Grain for Green Project follows a similar trajectory, topographic data are used to eliminate such cases, as the project typically applies to farmland with a slope greater than 25° (S1 Fig). The specific mapping method and processing steps are shown in S1 Text. In addition, the abandoned farmland rate ($FAR_i$) is defined as follows:

$$FAR_i = \frac{FA_i}{A_i} * 100\%$$

where $FA_i$ represents the abandoned farmland area in the year of i, $A_i$ represents the farmland area in the year of i.

**Modeling of the willingness to farmland abandonment.**

(1) **Logit model**

Taking elderly households as an example, whether farmers abandon farmland is a binary, discrete variable. We applied a logit model to estimate the impact of internal and external factors on elderly households' willingness to abandon farmland [59,60]. The model can be expressed using the following equation:

$$Logit\left(FAp_{it}\right) = \beta_0 + \beta_1 region\_age_{it} + \beta_2 household\_age_{it} + \sum_{a=1}^{m} \beta_{3a}F_{ita} + \sum_{b=1}^{n} \beta_{3b}U_{itb} + \sum_{c=1}^{k} \beta_{3c}E_{itc} + \theta_t + \varepsilon_{it1}$$

In this expression, $FAp_{it}$ denotes the probability of the $i_{th}$ elderly households' farmland is abandoned, $Logit\left(FAp_{it}\right)$ is an abandonment dummy variable of $i_{th}$ elderly household (abandonment = 1, non-abandonment = 0); $region\_age_{it}$ is the variables representing the rural aging in the region, $household\_age_{it}$ is the variables representing the households' age; $F_{ita}$ is a set of farmer features variables apart from $household\_age_{it}$; $U_{itb}$ is a set of farmland use features variables; $E_{itc}$ is a set of environmental features variables; $\theta_t$ is time dummy variable; $\beta_0$ is the regression intercept and $\varepsilon_{it}$ is the random disturbance term.

(2) **Variables selection**

**Dependent variable.** To reflect the problem of farmland abandonment, the dependent variable in this study is whether farmland is abandoned. In this study, the questionnaire asked, "Do households abandon farmland?". The answers to this question are assigned a value of "1" for abandonment and "0" for non-abandonment.

**Key explanatory variables.** The key explanatory variable is the degree of rural population aging, which is measured at both regional and household levels. At the regional level, the age structure of the resident population was used to reflect rural population aging [60], specifically focusing on the proportion of rural residents aged 60–69 (*PPA60_69*) and the proportion of rural residents aged 70 and over (*PPA70*) [61]. Meanwhile, at the household level, this study measures rural population aging as the "average age of farming households" (*age*) and "the proportion of the farmers aged 60 years or older in the total number of farmers in the family (*old farmer*)" [62].

**Other explanatory variables.** Previous studies illustrated that the farmland abandonment behavior of households is mainly under impact of 3 types of variables: farmer features, the utilization characteristics of farmland, and local socio-economic and natural environment [4,12,63]. By referring to the literature concerning the causes of farmland abandonment and the second chapter of this study at the farmer level, 4 factors are selected in terms of farmer features, including *gender*, *education*, *farm laborers*, and *household type*. 6 factors are selected in terms of farmland use features, including *farmland area per capita, proportion of dryland, land quantity, distance to home, contract, and sublet*. 3 factors are selected in terms of environmental features, including *landform type, village type, and distance to county*. What is listed below in Table 2 is the descriptive statistics concerning these variables.

**Table 2. Variable settings and assignment descriptions.**

| Variables | Definition | Elderly farmers | | Young farmers | |
|---|---|---|---|---|---|
| | | Mean | S.D. | Mean | S.D. |
| FA | Whether the farmer abandons farmland (1 = yes; 0 = no) | 0.196 | 0.398 | 0.153 | 0.360 |
| PPA60–69 | Proportion of population aged 60–69 | 0.275 | 0.060 | 0.273 | 0.059 |
| PPA70 | Proportion of population aged 70 and over | 0.199 | 0.047 | 0.195 | 0.047 |
| Age | Average age of farming households | 67.390 | 4.708 | 53.446 | 5.227 |
| Old farmer | The proportion of farmers aged 60 or above in the total number of family farmers | 0.968 | 0.126 | 0.127 | 0.233 |
| Gender | The gender of the household head | 1.190 | 0.393 | 1.185 | 0.389 |
| Education | The education level of the household head (1 = no education, 2 = primary school education, 3 = junior high school education, 4 = senior high school education, 5 = college education and above) | 2.082 | 0.874 | 2.461 | 0.850 |
| Farm laborers | The number of the agricultural labor force | 1.639 | 0.501 | 1.727 | 0.598 |
| Household type | The type of the household (1 = agriculture-led household, 2 = agro-pastoral household; 3 = self-sufficient household) | 1.774 | 0.847 | 1.603 | 0.708 |
| Farmland area per capita | The area of farmland per person in a household | 19.361 | 74.016 | 51.204 | 125.256 |
| Proportion of dryland | The proportion of dryland to total area of farmland | 0.833 | 0.342 | 0.678 | 0.440 |
| Land quantity | The number of farmland parcels | 4.637 | 2.870 | 5.537 | 3.884 |
| Distance to home | The average distance from the plots to home | 1.724 | 1.685 | 1.865 | 2.227 |
| Contract | Whether the investigated farmer has contracted farmland from others (1 = yes; 0 = no) | 0.171 | 0.377 | 0.394 | 0.489 |
| Sublet | The investigated farmer has subleased farmland to others (1 = yes; 0 = no) | 0.299 | 0.458 | 0.209 | 0.407 |
| Landform type | The landform type of the plot (1 = flat; 2 = sloping) | 1.392 | 0.488 | 1.337 | 0.473 |
| Village type | Administrative types of villages (1 = natural village; 2 = administrative) | 1.413 | 0.493 | 1.483 | 0.500 |
| Distance to county | The distance from the village to the county | 20.704 | 16.238 | 20.075 | 15.077 |

Note: S.D. is the standard deviation.

**Machine learning model for predicting future farmland abandonment.** The future population structure data of Inner Mongolia under different shared socio-economic path scenarios (SSPs) were used to reveal changes in farmers' farmland abandonment in the future. Specifically, SSP1, SSP2, and SSP5 scenarios were selected to obtain the proportion of the population aged 60–69 and aged 70 and above in the total population in 2100 [56]. The random forest (RF) model was then used to predict future farmland abandonment behavior of farmers. SSP1, SSP2, and SSP5 respectively describe a sustainable development scenario, a business-as-usual scenario, and a fossil-fueled development scenario [64].

The random forest is a representative of ensemble learning, which primarily operates by constructing multiple decision trees and synthesizing their prediction results [65]. The parameters of the random forest model used in this study include 500 decision trees (Ntree), 7 variables per decision tree (Mtry), and a minimum node (nodesize) set to 1. Based on this, it predicts whether farmers will abandon farmland in the future according to the principle of the minority obeying the majority [66]. Considering that the classification result represents the probability of occurrence of binary categories, the overall accuracy, precision, and recall are used as evaluation metrics for the random forest model's accuracy [67] (S2 Text).

**Ethics statement.** This study did not require ethics approval, as it involved non-invasive field surveys conducted at the village level. For field research involving questionnaires, we first contacted the village secretary to explain the research purpose and gather basic village information. Upon receiving their permission, we interacted with individual farmers. Before conducting the survey, we identified ourselves to the farmers and explained the survey's purpose, the intended use of the data, and the confidentiality measures in place. All questionnaires received oral consent from each household

before proceeding with the survey. No personally identifiable information was collected during the study and all data were analyzed anonymously to ensure participant privacy.

## Results

### Spatio-temporal changes of abandoned farmland

From 2000 to 2020, the farmland in the Ulanqab decreased from 9734 km$^2$ to 7464 km$^2$, resulting in a net reduction of 2270 km$^2$. Meanwhile, the region exhibited a significantly increasing trend of grassland from 43612 km$^2$ to 45383 km$^2$, with a net gain of 1771 km$^2$ (Fig 5a). Notably, although the absolute area of abandoned farmland declined slightly from 295 km$^2$ in 2000 to 273 km$^2$ in 2020, the abandonment rate increased slightly from 3.03% to 3.66%.

Specifically, the temporal evolution of farmland abandonment can be broadly divided into three phases. The first phase (2000–2006) was the peak period of abandonment. The area of farmland abandonment reached a historical high of approximately 562 km$^2$ in 2002, with a corresponding abandonment rate of 6.12%. In 2005, the abandonment rate reached its peak of 6.47%. The second phase (2007–2012) showed an overall downward trend. Both the abandonment area and rate declined and remained at relatively low levels during this period. The lowest values were observed in 2011, with 194 km$^2$ of abandoned farmland and an abandonment rate of 2.42%. The third phase (2013–2020) remained at a

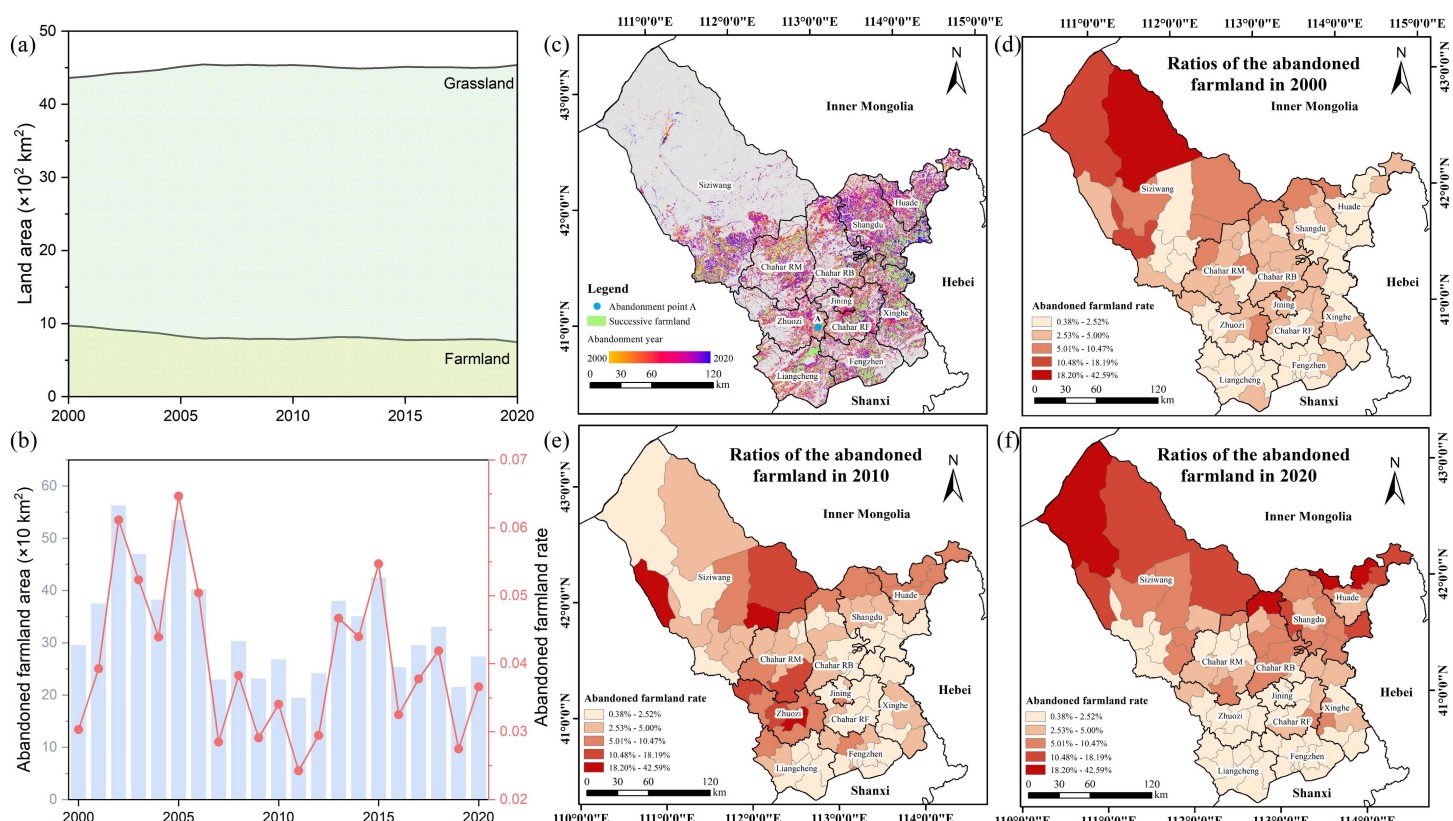

**Fig 5. Spatiotemporal pattern of abandonment from 2000 to 2020. (a)** Temporal variation in grassland area and farmland area. **(b)** Temporal variation in abandoned farmland area and the abandonment rate. **(c)** Spatial distribution of abandoned farmland. **(d-f)** Ratios of the abandoned farmland from 2000 to 2020. All map boundary data in this figure are consistent and publicly available from the National Platform for Common Geospatial Information Services (www.tianditu.gov.cn). The review map number is GS (2024) 0650.

moderate level but a slight decline. Although the abandoned area and abandonment rate rebounded in 2015, with 472 km$^2$ of abandoned farmland and an abandonment rate of 5.47%, the subsequent continuous downward trend was significant, and by 2019, the abandoned area and abandonment rate had dropped to 239 km$^2$ and 2.75%.

In general, the temporal and spatial characteristics of abandoned farmland in Ulanqab exhibited significant changes from 2000 to 2020 (Fig 5c). In 2000, the majority of abandoned farmland was concentrated in the northern part of Siziwang Banner (Fig 5d). Meanwhile, the abandonment rate gradually increased, particularly in Zhuozi, the southern part of Chahar Right Middle Banner, and the eastern part of Siziwang Banner by 2010 (Fig 5e). In 2020, areas with a high rate of farmland abandonment were predominantly located in the northern region, particularly in Siziwang Banner, Chahar Right Back Banner, Shangdu and the northern part of Huade, while areas with a low rate of farmland abandonment were predominantly distributed in the southern region, dominated by the Jining, Fengzhen, Zhuozi and Liangcheng, with an abandonment rate of less than 2.52% (Fig 5f).

**Factors influencing farmers' willingness to abandon farmland in Ulanqab**

**Basic information of households of different age groups.** The basic information of households of different age groups is shown in Table 3. Analysis of household characteristics showed that 45.8% of households are mildly aged households, and their members have become the primary labor force in rural areas, accounting for 46.6%. Furthermore, mildly aged households and severely aged households exhibited lower education levels and fewer agricultural labor forces compared to youth households. As households aged, a tendency toward self-sufficient farming was observed, with 22.5% of mildly aged households and 37.6% of severely aged households engaging in it (Table 3).

**Comparative analysis of factors affecting farmland abandonment among households of different age groups.** The results of model estimation are shown in Table 4. A variance inflation factor (VIF) analysis was conducted to assess the multicollinearity between the variables. The tolerance values of both models were less than 1, and the VIF was well below 10, with a minimum of 1.040 and a maximum of 7.349, indicating the absence of multicollinearity (Table 4). The

Table 3. The attribute of households of different age groups.

| Category | Value | Youth households | | Middle-aged households | | Mildly aged households | | Severely aged households | |
|---|---|---|---|---|---|---|---|---|---|
| | | Number (35) | Proportion (2.9%) | Number (371) | Proportion (31.0%) | Number (547) | Proportion (45.8%) | Number (242) | Proportion (20.3%) |
| Gender | Male | 29 | 82.9 | 302 | 81.4 | 438 | 80.1 | 201 | 83.1 |
| | Female | 6 | 17.1 | 69 | 18.6 | 109 | 19.9 | 41 | 16.9 |
| Education | Illiteracy | 4 | 11.4 | 48 | 12.9 | 157 | 28.7 | 63 | 26.0 |
| | Primary school | 12 | 34.3 | 141 | 38.0 | 215 | 39.3 | 117 | 48.4 |
| | Junior high school | 16 | 45.7 | 154 | 41.5 | 141 | 25.8 | 51 | 21.1 |
| | Senior high school and above/ Polytechnic school and above | 3 | 8.6 | 28 | 7.6 | 34 | 6.2 | 11 | 4.5 |
| Farm laborers | One | 19 | 54.3 | 123 | 33.2 | 176 | 32.2 | 117 | 48.3 |
| | Two | 14 | 40.0 | 227 | 61.2 | 364 | 66.5 | 124 | 51.3 |
| | Three or more | 2 | 5.7 | 21 | 5.6 | 7 | 1.3 | 1 | 0.4 |
| Household type | Self-sufficient household | 2 | 5.7 | 51 | 13.7 | 123 | 22.5 | 91 | 37.6 |
| | Non-self-sufficient household | 33 | 94.3 | 320 | 86.3 | 424 | 77.5 | 151 | 62.4 |
| | Agriculture-led household | 21 | 60.0 | 193 | 52.0 | 283 | 51.7 | 109 | 45.0 |
| | Agro-pastoral household | 12 | 34.3 | 127 | 34.3 | 141 | 25.8 | 42 | 17.4 |

Note: household type is first divided into self-sufficient household and non-self-sufficient household, and then non-self-sufficient household is divided into agriculture-led household and agro-pastoral household.

**Table 4. Logit regression results of the impact of explanatory variables on different age households' willingness to abandon farmland.**

| Variables | Elderly households | | | Young households | | |
|---|---|---|---|---|---|---|
| | TOL | VIF | Logit | TOL | VIF | Logit |
| PPA60–69 | 0.138 | 7.234 | −21.65*** (5.432) | 0.150 | 6.669 | −17.81** (7.523) |
| PPA70 | 0.136 | 7.349 | 24.53*** (6.851) | 0.151 | 6.632 | 25.90*** (9.586) |
| Age | 0.859 | 1.164 | 0.041* (0.023) | 0.753 | 1.327 | −0.0175 (0.0359) |
| Old farmer | 0.863 | 1.158 | −0.587 (0.843) | 0.683 | 1.465 | 0.205 (0.792) |
| Gender (2) | 0.874 | 1.144 | 0.025 (0.303) | 0.842 | 1.188 | −0.945* (0.508) |
| Education (2) | 0.853 | 1.172 | 0.206 (0.262) | 0.854 | 1.170 | 0.277 (0.563) |
| Education (3) | 0.853 | 1.172 | 0.251 (0.305) | 0.854 | 1.170 | 0.145 (0.567) |
| Education (4) | 0.853 | 1.172 | 0.680 (0.475) | 0.854 | 1.170 | −1.259 (0.983) |
| Education (5) | 0.853 | 1.172 | 2.911** (1.322) | 0.854 | 1.170 | 0.434 (1.436) |
| Farm laborers | 0.914 | 1.094 | −0.472** (0.237) | 0.756 | 1.324 | −0.320 (0.329) |
| Household type (2) | 0.950 | 1.053 | −0.212 (0.285) | 0.888 | 1.127 | 0.251 (0.380) |
| Household type (3) | 0.950 | 1.053 | 0.229 (0.245) | 0.888 | 1.127 | 1.016** (0.495) |
| Farmland area per capita | 0.908 | 1.102 | −0.042*** (0.012) | 0.813 | 1.230 | −0.0125 (0.00883) |
| Proportion of dryland | 0.854 | 1.171 | 0.809** (0.352) | 0.709 | 1.410 | 0.129 (0.489) |
| Land quantity | 0.921 | 1.085 | 0.198*** (0.040) | 0.887 | 1.127 | 0.118** (0.0527) |
| Distance to home | 0.896 | 1.116 | −0.059 (0.071) | 0.920 | 1.087 | 0.125* (0.0738) |
| Contract | 0.909 | 1.100 | −0.756** (0.355) | 0.814 | 1.228 | −1.217*** (0.427) |
| Sublet | 0.885 | 1.129 | −0.016 (0.230) | 0.899 | 1.112 | −0.174 (0.416) |
| Landform type (2) | 0.907 | 1.102 | 0.107 (0.228) | 0.809 | 1.236 | 0.831** (0.364) |
| Village type (2) | 0.823 | 1.215 | 0.194 (0.214) | 0.949 | 1.054 | 0.382 (0.333) |
| Distance to county | 0.961 | 1.040 | −0.0003 (0.008) | 0.686 | 1.457 | −0.00757 (0.0147) |
| Constant | – | – | −1.517 (1.746) | – | – | −0.174 (2.300) |
| Year dummies | – | – | Yes | – | – | Yes |
| Log likelihood | – | – | −313.21 | – | – | −131.15 |
| LR chi2 | – | – | 155.37 | – | – | 84.72 |
| N | – | – | 789 | – | – | 406 |

Note. ***, **, * denotes the significant statistical level of 0.01, 0.05, and 0.1. The number in brackets after the variable is the value of the dummy variable. Standard errors are in parentheses below the correlation coefficients.

LR chi2 for both models were significant, indicating that the models fit the data well, and the estimates of the two models are valid。

The estimation results indicated that *PPA60_69, PPA70, age, education (5), farm laborers, farmland area per capita, proportion of dryland, land quantity,* and *contract* had significant impacts on elderly households' willingness to abandon farmland. Meanwhile, *PPA60_69, PPA70_79, gender, household type (3), land quantity, distance to home, contract*, and *landform type (2)* had significant impacts on young households' willingness to abandon farmland (Table 4). Therefore, we can confirm Hypothesis 2 that the driving factors of farmland abandonment differ significantly across different age groups of households.

At the regional level, both models showed a significant negative correlation between the proportion of rural residents aged 60–69 years and farmland abandonment. Conversely, the proportion of rural residents aged 70 and above exhibited a significant positive correlation with farmland abandonment, suggesting that an increase in the severely aged population will exacerbate farmland abandonment. Therefore, we can confirm Hypothesis 3 that an increase in the proportion of the mildly aged population will alleviate farmland abandonment to some extent.

At the household level, age, as the key factor, has a positive impact on abandonment among the elderly households. For each additional year of age for elderly households, the probability of farmland abandonment increases by 4.1%. Therefore, we can confirm Hypothesis 1 that age will exacerbate farmland abandonment in elderly households. Nevertheless, it is noteworthy that among young households, age has a negative impact on farmland abandonment, though the results are not significant.

For elderly households, both *education (5), the proportion of dryland,* and *land quantity* exhibited a significant positive impact on elderly households' willingness to abandon farmland, with partial correlation coefficients of 2.911, 0.809, and 0.198, respectively. As household education levels increase, the likelihood of opting for alternative income sources rises, leading to farmland abandonment. Additionally, cultivating drylands is economically inefficient. Therefore, a higher proportion of drylands in a household, the more likely the household will choose to abandon part of drylands to maximize their benefits. It was easy to understand that households with more farmland parcels had a higher willingness to abandon farmland. A possible reason for this was that a larger number of farmland parcels required households to expend more energy in cultivation, and consequently, some labor-deficient households may choose to abandon marginal plots, particularly those farther away, to alleviate the pressure of farming. Conversely, *farm laborers, farmland area per capita* and *contract* were negative indicators affecting elderly households' willingness to abandon farmland, with partial correlation coefficients of −0.472, −0.042, and −0.756, respectively. As the number of labor forces declines, households were unable to maintain the original area of farmland, resulting in the decision to abandon. A negative correlation coefficient of farmland area per capita and contract status further confirms that farmland abandonment was closely related to the households' management ability. Households with higher per capita farmland and contracted farmland exhibited characteristics of scale in farming machinery and management facilities, indicating stronger management ability, therefore, they showed a lower willingness to abandon farmland.

## Changes in population and farmland abandonment from 2020 to 2100

From 2010 to 2100, differences in the population age structure were observed under different scenarios. In the SSP1 and SSP5 scenarios, the age structure of the population in 2100 is nearly identical, with slight differences in the proportion of population over 70 years old, at 38.91% and 38.89%, respectively. In the SSP2 scenario, the proportion of population aged 60–69 is 11.35%, and the proportion of those aged 70 and above is 27.29%, both of which are lower than in the other two scenarios (Fig 6a). Moreover, the population of Inner Mongolia presented a trend of first rising and then declining under different scenarios (Fig 6b).

The proportion of farmers adopting abandonment behavior in Ulanqab in 2100 was predicted based on the random forest model, which achieved an accuracy of 82.85%, a recall rate of 97.47%, and a precision of 84.30%. The results are

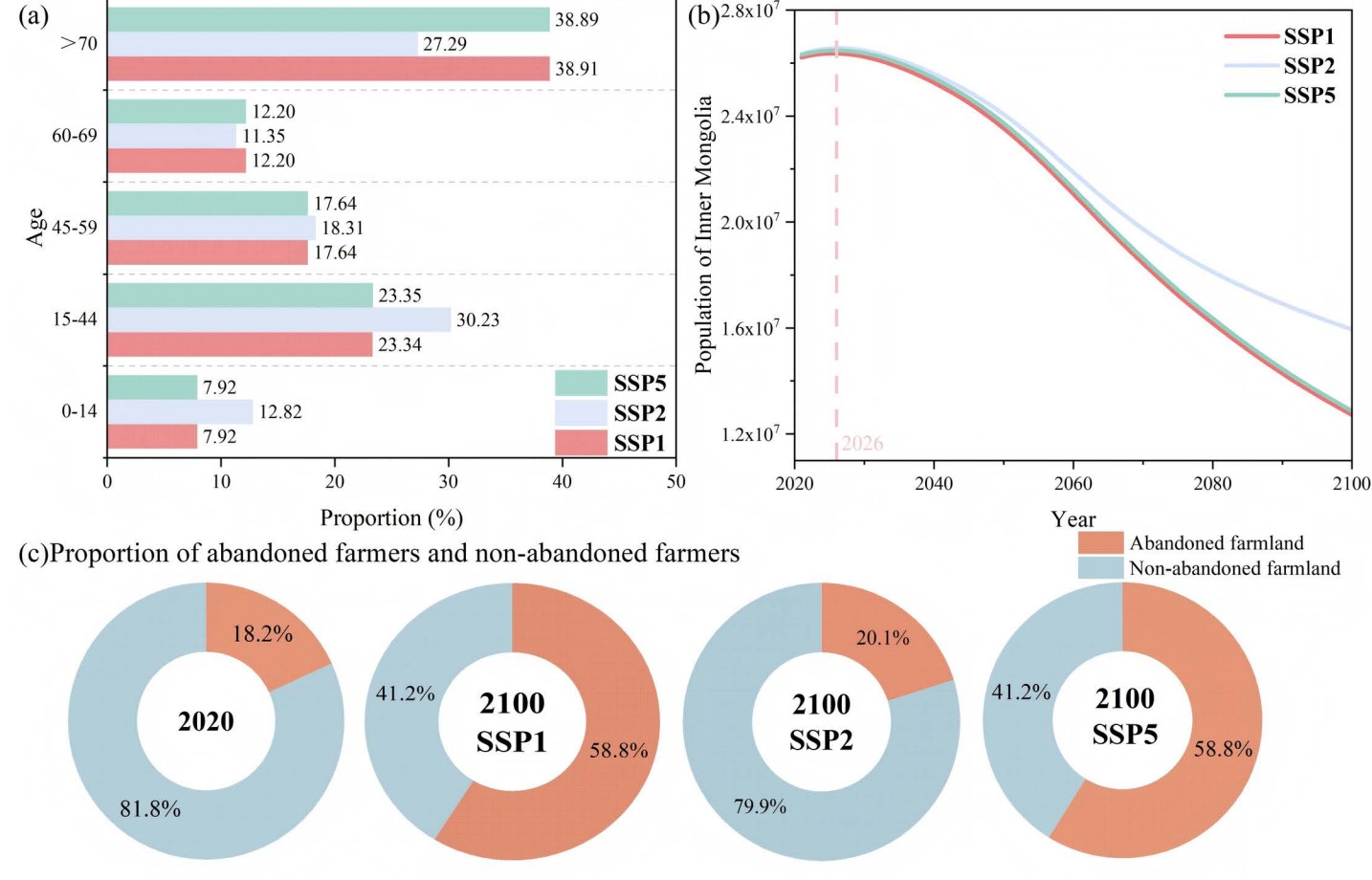

**Fig 6. Changes in farmland abandonment based on SSPs Scenarios. (a)** Future population age structure in Inner Mongolia. **(b)** Future population changes from 2021 to 2100 in Inner Mongolia. **(c)** Proportion of abandoned and non-abandoned farmers in Ulanqab in 2100.

shown in Fig 6c. In the SSP1 and SSP5 scenarios, compared to the questionnaire data, the proportion of the population aged 60–69 decreased, while the proportion of those aged 70 and above increased. The proportion of farmers engaging in abandonment behavior increased significantly, reaching 58.8%. These scenarios suggested that a decreasing of the mildly aged population and an increasing of the severely aged population is likely to exacerbate farmland abandonment. In contrast, in the SSP2 scenario, the increase in the proportion of the population aged 70 and above was lower than in the other two scenarios, with the abandonment proportion at 20.1%.

## Mechanism of rural population aging on farmland abandonment

At the regional level, a random forest model was further employed to simulate scenarios where the proportion of the mildly aged population (60–69) remained unchanged, while the proportion of the severely elderly population increased by 5% and 10%, respectively. The results revealed that the abandonment rate of farmers increased to 24.4% and 34.0%, respectively. These findings suggested that as the severely aged population increases, the agricultural labor supply becomes increasingly constrained, thereby intensifying the risk of land abandonment. Conversely, additional simulations were conducted under the condition that the proportion of the mildly aged population increased by 5% and 10%, while

the proportion of the severely aged population remained unchanged. The abandonment rate of farmers was found to decrease to 11.4% and 10.5%, respectively. These findings provided a favorable explanation for the increasing proportion of the mildly aged population mitigates farmland abandonment in the FPENC.

At the household level, population aging is reflected in the increase in the average age of family members and the decrease in the number of agricultural laborers. Empirical evidence confirms that the average age of households and the number of agricultural laborers were key determinants of farmland abandonment highlighting the complex impact mechanism of rural population aging on farmland abandonment.

## Discussion

### Driving factors of farmland abandonment in Ulanqab

Identifying the driving factors influencing farmland abandonment is a complex process. Based on the regional characteristics of deep rural population aging, this study focuses on analyzing the impact of rural population aging on farmland abandonment from both regional and household perspectives, thereby effectively supplementing the limited attention currently given to farmland abandonment in relation to changes in population age structure [4,68]. Moreover, previous studies have identified various determinants of abandonment, including land type, land quality, number of farm laborers, the average age of farm laborers and farmland area per capita [4,17,18,36,38], which is consistent with our research findings.

In addition, the extent to which aging affects farmland abandonment is constrained by natural and socio-economic conditions. The living material environment, as an exogenous factor, also plays a significant role in farmland abandonment. The results suggest that farmlands with poor quality conditions, and slope landform, not only faced difficulties in cultivation but also brought limited benefits. Consequently, households are more prone to abandon these farmlands under the same conditions [16,30,69]. Moreover, a noteworthy driver is farmland transfer, which can reduce the probability of farmland abandonment [70]. Therefore, in the future, more thoroughly natural and socio-economic factors should be integrated to explore the compound driving mechanisms of farmland abandonment under the context of population aging and then provide a scientific foundation for more targeted policy interventions and the optimization of land use strategies.

### Innovations and uncertainties

Remote sensing data and survey data were employed to comprehensively assess the mechanism of rural population aging on farmland abandonment in the FPENC. One source of uncertainty, however, is that the accuracy of abandoned farmland maps obtained by the trajectory-based change detection approach is determined according to the accuracy of the published LULC products. Compared with other datasets, the CLCD data offers advantages in temporal and spatial resolution [54]. However, the presence of mixed pixels and the spectral similarity between farmland and grassland may lead to errors and misclassification in the analysis of farmland dynamics [71]. Consequently, the results were further revised using the CACD data. Further research on identifying abandoned farmland with more accurate land use data would be worthwhile.

Additionally, the age structure of the population under different SSP scenarios is limited to the provincial level, and do not distinguish between rural and urban areas [56]. Meanwhile, significant differences in the age structure across regions of Inner Mongolia indicate that provincial-level data may not accurately represent the situation in Ulanqab. Therefore, further research is required to refine regional corrections of population data in future scenarios to enhance estimation accuracy. Notwithstanding these limitations, the study added to our understanding of the impact of rural population aging on farmland abandonment.

### Policy implications

This study deeply analyzed the effect of rural population aging on farmland abandonment. Similar to many countries, China is experiencing a rapidly aging rural population [22]. Considering that China's overall aging process lags behind that

of the FPENC, understanding farmland abandonment within the context of rural population aging in the FPENC provides valuable insights for China's future responses to this issue. Ulanqab has suboptimal agricultural production conditions, including severe drought, water scarcity, and a fragile ecological environment. These natural constraints, coupled with the pronounced aging of the rural population, contribute to a more complex and severe farmland abandonment issue compared to other regions. Consequently, developing effective strategies for managing abandoned farmland in such areas holds significant practical relevance. In addressing these challenges, it is essential to consider not only food security concerns but also the unique ecological functions that abandoned farmland may serve [72,73].

The results showed that 45.8% of the surveyed households were classified as mildly aged households, whose members have become the primary labor force in rural areas, accounting for 46.6%. Meanwhile, according to the seventh census data, the mildly aged population (60–69) in rural areas of Ulanqab constituted 58% of the total rural aging population (≥60) in 2020. Compared with youth households and severely aged households, mildly aged households are less likely to abandon their farmland. Therefore, in the short term, an increase in the proportion of the mildly aged population in rural areas will contribute a certain amount of low-quality labor, thereby mitigating farmland abandonment and addressing the pressing concern of "who will farm the land".

In the long term, agricultural scaling and modernization represent essential strategies for mitigating farmland abandonment. Specifically, promoting mechanization and encouraging the adoption of modern agricultural technologies by households could enhance farmland utilization and agricultural productivity, thereby reducing farmland abandonment. Simultaneously, the potential of younger populations should be fully leveraged, and their critical role in sustaining agricultural heritage should be emphasized. In areas with severe aging and fragile ecological environment, eco-friendly agricultural practices should be promoted to meet the needs of severely aged households and protect the ecological environment.

Furthermore, the Chinese government must pay much attention to the construction of ecological civilization in ecologically fragile areas. Farmland abandonment can significantly influence the ecological environment, encompassing both positive aspects [74] and negative aspects [10,75]. Consequently, the utilization of abandoned farmland should consider a balanced approach between regional production and ecological sustainability. For farmland that has already been abandoned with low yield and poor quality, integrating it into the Grain for Green Project could serve as a viable strategy. Conversely, abandoned farmland with high-quality conditions should be prioritized for restoration to cultivation to enhance regional food security.

## Conclusion

Here we presented a study that analyzed the impact of rural population aging on farmland abandonment in the FPENC. The dynamic changes in farmland abandonment in the Ulanqab from 2000 to 2020 were systematically explored using a trajectory-based land use change detection approach. Binary logit regression models were then used to determine the willingness of both elderly and young households to abandon farmland based on field survey questionnaire data, and a random forest model was employed to predict the future trend of farmland abandonment under different SSPs. We observed an overall downward trend of farmland abandonment from 295 km$^2$ to 273 km$^2$ over the past 20 years. However, the abandonment rate increased slightly from 3.03% to 3.66%, with higher abandonment rates concentrated in the northern part of Ulanqab. Analysis of the questionnaire data showed that members of mildly aged households have become the primary labor force in rural agricultural production. Regression model results suggested that an increase in the proportion of the mildly aged population mitigated farmland abandonment, while an increase in the proportion of the severely aged population exacerbated it. The mechanisms underlying abandonment behavior differed between young and elderly households. For elderly households, the probability of farmland abandonment increased by approximately 4.1% for each additional year of age. Moreover, education, the proportion of dryland, land quantity, farm laborers, farmland area per capita, and whether farmland is contracted were also important factors influencing elderly households' willingness to abandon farmland. Random forest prediction results indicated that the proportion of abandoned farmland were projected

to increase due to changes in population age structure under various population aging scenarios by 2100. We also found that while the proportion of the severely aged population remained unchanged, an increase of 5% and 10% in the proportion of the mildly aged population could decrease the abandonment rate of farmers to 11.4% and 10.5%, respectively. This study provides insights into farmland abandonment in FPENC from a demographic perspective. Meanwhile, it highlights the essential role of the mildly aged population in mitigating farmland abandonment in the future amidst the ongoing trend of rural population aging.

## Supporting information

**S1 Text. Identification process of farmland abandonment.**
(DOCX)

**S1 Fig. Identification process of farmland abandonment.** All map boundary data in this figure are consistent and publicly available from the National Platform for Common Geospatial Information Services (www.tianditu.gov.cn). The review map number is GS (2024) 0650.
(TIF)

**S2 Text. Accuracy verification method of random forest.**
(DOCX)

## Acknowledgments

We would like to thank the Ulanqab Government for their assistance with statistical data and the questionnaires collection.

## Author contributions

**Conceptualization:** Yuling Jin, Pingli An.

**Funding acquisition:** Pingli An.

**Investigation:** Yukai Wei, Sicheng Mao, Haile Zhao, Wenting Liu.

**Methodology:** Yuling Jin, Yi Zhou.

**Resources:** Yukai Wei, Zhihua Pan.

**Software:** Yuling Jin, Yi Zhou.

**Supervision:** Guoliang Zhang, Zhihua Pan, Pingli An.

**Validation:** Yuling Jin, Sicheng Mao, Haile Zhao, Wenting Liu.

**Visualization:** Yuling Jin.

**Writing – original draft:** Yuling Jin.

**Writing – review & editing:** Guoliang Zhang, Xin Chen.

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
