## [Decision Letter · Decision Letter 0]

PONE-D-25-04045Increasing proportion of mildly aged population in rural mitigates farmland abandonment in the farming-pastoral ecotone of northern ChinaPLOS ONE

Dear Dr. An,

Thank you for submitting your manuscript to PLOS ONE. After careful consideration, we feel that it has merit but does not fully meet PLOS ONE’s publication criteria as it currently stands. Therefore, we invite you to submit a revised version of the manuscript that addresses the points raised during the review process.

Now I have received two valid comments. Please make systematic revisions to the manuscript based on the reviewers' opinions and provide one-on-one responses.

We look forward to receiving your revised manuscript.

Kind regards,

Dingde Xu

Academic Editor

PLOS ONE

Journal Requirements:

“This work was supported by the National Natural Science Foundation of China

(grant number 42271268)”

5. We note that Figures 3-5 in your submission contain map/satellite images which may be copyrighted. All PLOS content is published under the Creative Commons Attribution License (CC BY 4.0), which means that the manuscript, images, and Supporting Information files will be freely available online, and any third party is permitted to access, download, copy, distribute, and use these materials in any way, even commercially, with proper attribution. For these reasons, we cannot publish previously copyrighted maps or satellite images created using proprietary data, such as Google software (Google Maps, Street View, and Earth). For more information, see our copyright guidelines: http://journals.plos.org/plosone/s/licenses-and-copyright.

 1. You may seek permission from the original copyright holder of Figure 3-5 to publish the content specifically under the CC BY 4.0 license. 

Additional Editor Comments:

Now I have received two valid comments. Please make systematic revisions to the manuscript based on the reviewers' opinions and provide one-on-one responses.

Reviewers' comments:

Reviewer's Responses to Questions

**Comments to the Author**

1. Is the manuscript technically sound, and do the data support the conclusions?

Reviewer #1: Yes

Reviewer #2: Yes

2. Has the statistical analysis been performed appropriately and rigorously? 

Reviewer #1: No

Reviewer #2: Yes

3. Have the authors made all data underlying the findings in their manuscript fully available?

Reviewer #1: No

Reviewer #2: Yes

4. Is the manuscript presented in an intelligible fashion and written in standard English?

Reviewer #1: Yes

Reviewer #2: Yes

5. Review Comments to the Author

Reviewer #1: This manuscript is of great significance to reveal the mechanism of influence of different aging degrees on farmland abandonment and to formulate countermeasures for farmland abandonment. The main questions are as follows:

1. When presenting “Spatio-temporal changes of abandoned farmland”, please tell use changes in the ratio of land abandonment, in addition to changes in the area of land abandonment.

2. I think there is no need to present the different behavioral patterns across different age groups in Lines 318-339, as it is not particularly relevant to the core content (land abandonment) of the manuscript.

3. “Mechanism of rural population aging on farmland abandonment” should be the content in Results.

4. I did not see the necessity of Table 6 in the manuscript.

5. All images have a lower resolution, which makes reading difficult.

Figure 1 is too complicated to understand. It is recommended to retain only the core elements.

Figure 7 is also not necessary.

Reviewer #2: Aging and the abandonment of farmland are unavoidable key issues faced by China's agricultural development. The author has conducted a relatively thorough analysis of the impact of aging on abandonment, especially by classifying the degree of family aging and exploring the heterogeneity of the impact of different degrees of aging on abandonment. Regarding the governance of abandonment in the context of aging, Thus, it has strong practical significance in ensuring food security and other aspects. However, the following deficiencies exist for your reference:

(1) Your topic is "The Increase in the proportion of lightly elderly people in Rural areas alleviates the abandonment of farmland in the Agro-pastoral Ecotone of Northern China". It should focus on this theme. Other aspects such as the prediction of abandonment are your auxiliary and supplementary analysis. Please grasp this key theme throughout the text.

(2) Abstract section and lines 321-323: Youdaoplaceholder0 aged households (60-69) were identified as the main agricultural labor force,accounting for 45.8%; “Analysis of household characteristics showed that the mildly aged households were the primary labor force in rural "areas, accounting for 45.8%" Your expression is incorrect. The statement that the family is the labor force is incorrect. Please revise.

(3) In your variable setting section, it seems that aging should be an explanatory variable rather than a key variable, and abandonment is also a key variable in this study.

(4) In your line 423: Youdaoplaceholder0, the aging population in rural areas serves as an endogenous driving force behind farmland abandonment. The description of endogenous motivation is inaccurate. It is suggested to be modified.

(5) Lines 471-473: Your policy recommendations should be based on your research conclusions. It is not advisable to add any other content here as the description is rather confusing. It is recommended to reorganize them.

(6) The introduction should highlight your research topic, namely two aspects: aging and abandonment. If the current background of aging and abandonment in China is not described clearly, the urgency and reality of the research should be emphasized. It is suggested to supplement relevant current background data. The introduction of the problem was rather abrupt

(7) In the section of theoretical analysis and research hypotheses, there are two issues: 1. The problem has not been thoroughly explained. Why does aging accelerate the abandonment of farmland and make it more likely for young families to abandon farmland? A limited theoretical exposition is not sufficient to arrive at your research hypothesis 123. The theoretical derivation of the assumption that there are differences within elderly families and that severely elderly people are more likely to accelerate abandonment is insufficient. It is suggested that the text be further polished.

6. PLOS authors have the option to publish the peer review history of their article (what does this mean? ). If published, this will include your full peer review and any attached files.

**Do you want your identity to be public for this peer review?** For information about this choice, including consent withdrawal, please see our Privacy Policy .

Reviewer #1: No

Reviewer #2: No

---

## [Author Response · Author response to Decision Letter 1]

17 Jun 2025

Dear Editor

PLOS One

Thank you for the opportunity to revise our Manuscript No. PONE-D-25-04045[Title: Increasing proportion of mildly aged population in rural mitigates farmland abandonment in the farming-pastoral ecotone of northern China]. We are grateful for the detailed comments and suggestions provided by each of the reviewers, and we have thoughtfully taken into account these comments.

We sincerely apologize for the inconvenience caused by the previous version of the manuscript. Due to our oversight, the uncorrected original manuscript was inadvertently submitted, resulting in numerous errors in the figures. We have now uploaded the corrected version and kindly ask you to review it at your convenience. We apologize for any confusion or disruption this may have caused. Additionally, we have added more contact information in the “Data Availability” file to support the long-term accessibility and oversight of the data.

Below, we provide a detailed response to each of the editorial comments and outline the revisions or explanations accordingly.

1. Format modification

We have modified the file format and prepared the following revised materials according to PLOS ONE requirements:

1)Response to Reviewers (this document)

2)Revised Manuscript with Track Changes

3)Clean Manuscript (without track changes)

4)Ethics Statement Document and Data availability (uploaded under “Other” files)

5)Cover letter (adding the funding statement)

2. Funding Statement

As requested, we have added the following standard disclaimer to the cover letter: “The funders had no role in study design, data collection and analysis, decision to publish, or preparation of the manuscript.”

3. Data Availability

We regret that we are unable to share the full dataset used in this study due to confidentiality agreements and the need to protect the geospatial privacy of local residents. These restrictions are in compliance with local data governance regulations.

However, all openly available data sources used in the study have been clearly listed in the manuscript. Further explanation regarding the data sharing constraints and the contact information for data requests has been added to the Data Availability statement.

4. Ethics Approval

We confirm that our study does not require ethics approval, as it does not involve human or animal subjects, and no personal or sensitive information was collected. To comply with the editorial request, we have obtained a formal explanation letter from our affiliated institution—College of Land Science and Technology, China Agricultural University.

The statement was reviewed and approved by:

1) Dr. Haishuang Jia, Director of Scientific Research and Social Services,

2) Ms. Zhaohong Cao, Deputy Party Secretary and Vice Dean,

3) Ms. Xiaohe Yu, Chief Officer of General Affairs.

We have uploaded the scanned letter, including the official stamp in English version, under “Other” files for your reference. Additionally, the original manuscript included the ethical statement within the questionnaire data section. To comply with the requirements, we have relocated the ethics statement to the “Methods” section of the manuscript, where it is now clearly and separately stated (lines 307–315).

5. Map and Satellite Image Copyright

We have addressed the copyright concerns in the following ways:

1)Figure 3: The original basemap has been removed.

2)Figures 3, 4, and 5: All geographic boundary data (including China’s administrative boundaries) were obtained from National Geomatics Center of China, a publicly accessible and free source. The satellite images in Figure 5b also have been removed.

We have updated the explanatory section following the image title to clearly indicate the data source and have included the map approval number GS (2024) 0650 in the manuscript.

6. Supporting information

We have revised and improved the Supporting Information section as requested. Titles have been added to all Supporting Information files and listed at the end of the manuscript.

7. Reference list

We have revised and improved the reference format in the main text and appendices as requested, and added or removed certain references based on the reviewer's suggestions and their relevance to the main content of the manuscript. The newly added references have been highlighted in red.

Due to revisions in the Introduction and Discussion sections, we have added three new references to enhance the theoretical foundation and contextual relevance of the manuscript. The details of the added references are as follows:

[2. Zheng QM, Ha T, Prishchepov AV, Zeng YW, Yin H, Koh LP. The neglected role of abandoned cropland in supporting both food security and climate change mitigation. Nature Communications. 2023;14(1):6083. doi: 10.1038/s41467-023-41837-y. (Line 561-563)

8. Li S, Li X. The mechanism of farmland marginalization in Chinese mountainous areas: Evidence from cost and return changes. Journal of Geographical Sciences. 2019; 29:531-48. (Line 577-578)

71. Ye J, Hu Y, Feng Z, Zhen L, Shi Y, Tian Q, et al. Monitoring of Cropland Abandonment and Land Reclamation in the Farming–Pastoral Zone of Northern China. Remote Sensing. 2024;16(6):1089. (Line 743-744)]

Considering the comments of reviewers, we have deleted certain sections of the manuscript and removed the corresponding references to maintain coherence and relevance.

[68. Figueiredo J, Pereira HM. Regime shifts in a socio-ecological model of farmland abandonment. Landscape Ecology. 2011;26(5):737-49. doi: 10.1007/s10980-011-9605-3. PubMed PMID: WOS:000291485100011. (in the original manuscript)

71. Han X, Wei C, Cao GY. Aging, generational shifts, and energy consumption in urban China. Proceedings of the National Academy of Sciences of the United States of America. 2022;119(37):e2210853119. Epub 2022/09/07. doi: 10.1073/pnas.2210853119. PubMed PMID: 36067298; PubMed Central PMCID: PMCPMC9478673. (in the original manuscript)

72. Deng X, Zeng M, Xu DD, Qi YB. Why do landslides impact farmland abandonment? Evidence from hilly and mountainous areas of rural China. Natural Hazards. 2022;113(1):699-718. doi: 10.1007/s11069-022-05320-z. PubMed PMID: WOS:000776420900003. (in the original manuscript)]

We also deleted a duplicate reference: [38. Wei Y, An P, Jin Y, Chen X, Zhang G, Pan Z. Population aging and its farmland effect on abandonment in the northern farming-pastoral ecotone: A case study of Ulanqab. Journal of Arid Land Resources and Environment. 2021;35(07):64-70. (in the original manuscript)]

8. Supplementary explanation

In response to the suggestions of editors and reviewers, we have made revisions in multiple sections of the manuscript. Additionally, during the revision process, we identified and corrected some grammatical and expression-related issues present in the original version. All modifications have been highlighted in red within the manuscript for ease of review.

Hopefully our responses and revisions are clear and meet the requirement of editor and reviewers. Moreover, please feel free to contact us if additional information is needed. We hope that all these changes fulfil the requirements to make the manuscript acceptable for publication in “PLOS One”.

Sincerely yours,

Dr. Pingli An

College of Land Science and Technology

China Agricultural University, China

List of Responses to Reviewers’ comment

Reviewer #1

Comment 1�When presenting “Spatio-temporal changes of abandoned farmland”, please tell use changes in the ratio of land abandonment, in addition to changes in the area of land abandonment.

[Answer: Thanks for your constructive comment and this is indeed something worth adding. we have revised and improved the manuscript by further supplemented the changes in the abandonment ratio within the section on spatio-temporal changes of abandoned farmland (Line 318-332). Additionally, due to copyright considerations, we have redrawn the original Fig. 5. The detailed revisions are as follows:

“Meanwhile, the region exhibited a significantly increasing trend of grassland from 436,12 km2 to 453,83 km2, with a net gain of 177.1 km2 (Fig 5a). Notably, although the absolute area of abandoned farmland declined slightly from 295 km2 in 2000 to 273 km2 in 2020, the abandonment rate increased slightly from 3.03% to 3.66%.

Specifically, the temporal evolution of farmland abandonment can be broadly divided into three phases. The first phase (2000-2006) was the peak period of abandonment. The area of farmland abandonment reached a historical high of approximately 562 km2 in 2002, with a corresponding abandonment rate of 6.12%. In 2005, the abandonment rate reached its peak of 6.47%. The second phase (2007-2012) showed an overall downward trend. Both the abandonment area and rate declined and remained at relatively low levels during this period. The lowest values were observed in 2011, with 194 km2 of abandoned farmland and an abandonment rate of 2.42%. The third phase (2013-2020) remained at a moderate level but a slight decline. Although the abandoned area and abandonment rate rebounded in 2015, with 472 km2 of abandoned farmland and an abandonment rate of 5.47%, the subsequent continuous downward trend was significant, and by 2019, the abandoned area and abandonment rate had dropped to 239 km2 and 2.75%.” (Line 318-332)

Fig 5. Spatiotemporal pattern of abandonment from 2000 to 2020 (Line 342)]

Comment 2�I think there is no need to present the different behavioral patterns across different age groups in Lines 318-339, as it is not particularly relevant to the core content (land abandonment) of the manuscript.

[Answer: Thank you very much for highlighting this issue. After careful consideration, we agree that the different behavioral patterns across different age groups was somewhat redundant, and we have therefore removed it from the manuscript. However, the section on basic household information remains relevant and important, as it reflects key characteristics of the surveyed households and also to some extent displays the relevant information of the questionnaire. To improve clarity and coherence, we have integrated this content into factors influencing farmers’ willingness to abandon farmland in Ulanqab Specifically, we have reorganized the section into two parts: (1) basic information of households of different age groups (Line 348), and (2) Comparative analysis of factors affecting farmland abandonment among households of different age groups (Line 359-360).]

Comment 3�“Mechanism of rural population aging on farmland abandonment” should be the content in Results.

[Answer: Thank you very much for pointing out it. We fully agree with your suggestion. Accordingly, and in consideration of the overall structure of the manuscript, we have made the following adjustments: we have relocated part of the original content titled “Mechanism of rural population aging on farmland abandonment” to the Results section, specifically in the second paragraph, to better emphasize the core theme of our research. The remaining content has been retained in the Discussion section but has been supplemented and reorganized to focus on other relevant factors influencing farmland abandonment in the context of population aging. Specifically, the revised content has been divided into two sections:

(1) Mechanism of rural population aging on farmland abandonment (Results, Line 432-449). The detailed revisions are as follows:

“At the regional level, a random forest model was further employed to simulate scenarios where the proportion of the mildly aged population (60-69) remained unchanged, while the proportion of the severely elderly population increased by 5% and 10%, respectively. The results revealed that the abandonment rate of farmers increased to 24.4% and 34.0%, respectively. These findings suggested that as the severely aged population increases, the agricultural labor supply becomes increasingly constrained, thereby intensifying the risk of land abandonment.” (Results, Line 434-439)

“At the household level, population aging is reflected in the increase in the average age of family members and the decrease in the number of agricultural laborers. Empirical evidence confirms that the average age of households and the number of agricultural laborers were key determinants of farmland abandonment highlighting the complex impact mechanism of rural population aging on farmland abandonment.” (Results, Line 445-449)

(2) Driving factors of farmland abandonment in Ulanqab (Discussion, Line 451-469). The detailed revisions are as follows:

“Identifying the driving factors influencing farmland abandonment is a complex process. Based on the regional characteristics of deep rural population aging, this study focuses on analyzing the impact of rural population aging on farmland abandonment from both regional and household perspectives, thereby effectively supplementing the limited attention currently given to farmland abandonment in relation to changes in population age structure [4, 68]. Moreover, Previous studies have identified various determinants of abandonment, including land type, land quality, number of farm laborers, the average age of farm laborers and farmland area per capita [4, 17, 18, 36, 38], which is consistent with our research findings.” (Discussion, Line 452-459)

“Therefore, in the future, more thoroughly natural and socio-economic factors should be integrated to explore the compound driving mechanisms of farmland abandonment under the context of population aging and then provide a scientific foundation for more targeted policy interventions and the optimization of land use strategies.” (Discussion, Line 466-469)]

Comment 4�I did not see the necessity of Table 6 in the manuscript.

[Answer: Thank you very much for your suggestion. After careful reconsideration, we also agree that Table 6 is somewhat redundant and does not substantially enhance the overall findings. Therefore, we have removed Table 6 from the manuscript to improve clarity and conciseness.]

Comment 5�All images have a lower resolution, which makes reading difficult. Figure 1 is too complicated to understand. It is recommended to retain only the core elements. Figure 7 is also not necessary.

[Answer: Many thanks for your suggestion. We sincerely apologize for the low resolution of the images and some of the images being too complex in our manuscript. In response, we have enhanced the resolution of all figures to ensure a clearer visual presentation. Meanwhile, we have simplified Figure 1 to improve its readability and supplemented missing elements in Figure 2. Additionally, after careful consideration, we concluded that Figure 7 does not contribute substantially to the manuscript and have therefore removed it. The detailed descriptions are as follows:

Fig. 1. Theoretical analysis framework for household farmland-use behavior (The theoretical framework and research hypotheses, Line 123)

Fig. 2. Household type and farmland abandonment mechanism. (The theoretical framework and research hypotheses, Line 161)]

Reviewer #2

Comment 1�Your topic is “The Increase in the proportion of lightly elderly people in Rural areas alleviates the abandonment of farmland in the Agro-pastoral Ecotone of Northern China”. It should focus on this theme. Other aspects such as the prediction of abandonment are your auxiliary and supplementary analysis. Please grasp this key theme throughout the text.

[Answer: Thank you very much for your suggestion. We deeply recognize that our initial description of the central theme was insufficient. Therefore, we further strengthen our grasp of the key theme of this manuscript. Specifically, for example, in “Changes in Population and Abandoned Farmland from 2020 to 2100” section, our original intention was to use future projections of farmland abandonment to reveal the distinct impacts of changes in the mildly and severely aged population structure. However, we acknowledge that our previous expression lacked precision and did not effectively highlight this objective. To address this, we have revised the section by removing irrelevant content and enhancing the emphasis on how variations in the proportions of mildly and severely aged populations affect farmland abandonment. These revisions aim to more clearly con

---

## [Decision Letter · Decision Letter 1]

Increasing proportion of mildly aged population in rural mitigates farmland abandonment in the farming-pastoral ecotone of northern China

PONE-D-25-04045R1

Dear Dr. An,

We’re pleased to inform you that your manuscript has been judged scientifically suitable for publication and will be formally accepted for publication once it meets all outstanding technical requirements.

Kind regards,

Dingde Xu

Academic Editor

PLOS ONE

Additional Editor Comments (optional):

The author has made revisions to the manuscript based on the reviewers' suggestions. Both reviewers have no further comments and recommend that it be directly accepted for publication.

Reviewers' comments:

Reviewer's Responses to Questions

**Comments to the Author**

1. If the authors have adequately addressed your comments raised in a previous round of review and you feel that this manuscript is now acceptable for publication, you may indicate that here to bypass the “Comments to the Author” section, enter your conflict of interest statement in the “Confidential to Editor” section, and submit your "Accept" recommendation.

Reviewer #1: All comments have been addressed

Reviewer #2: All comments have been addressed

2. Is the manuscript technically sound, and do the data support the conclusions?

Reviewer #1: Yes

Reviewer #2: Yes

3. Has the statistical analysis been performed appropriately and rigorously? 

Reviewer #1: Yes

Reviewer #2: Yes

4. Have the authors made all data underlying the findings in their manuscript fully available?

Reviewer #1: Yes

Reviewer #2: Yes

5. Is the manuscript presented in an intelligible fashion and written in standard English?

Reviewer #1: Yes

Reviewer #2: Yes

6. Review Comments to the Author

Reviewer #1: After a thorough review of the revised manuscript and a reading of the authors' responses, I think that all the comments have been addressed and the paper can be published in its current form.

Reviewer #2: (No Response)

7. PLOS authors have the option to publish the peer review history of their article (what does this mean? ). If published, this will include your full peer review and any attached files.

**Do you want your identity to be public for this peer review?** For information about this choice, including consent withdrawal, please see our Privacy Policy .

Reviewer #1: No

Reviewer #2: No

---

## [Editor Report · Acceptance letter]

PONE-D-25-04045R1

PLOS ONE

Dear Dr. An,

I'm pleased to inform you that your manuscript has been deemed suitable for publication in PLOS ONE. Congratulations! Your manuscript is now being handed over to our production team.

Kind regards,

on behalf of

Dr. Dingde Xu

Academic Editor

PLOS ONE